# Evaluation of NaCl and KCl Salting Effects on Technological Properties of Pre- and Post-Rigor Chicken Breasts at Various Ionic Strengths

**DOI:** 10.3390/foods9060721

**Published:** 2020-06-02

**Authors:** Dong-Heon Song, Youn-Kyung Ham, Sin-Woo Noh, Koo Bok Chin, Hyun-Wook Kim

**Affiliations:** 1Department of Animal Science and Biotechnology, Gyeongnam National University of Science and Technology, Jinju 52725, Korea; timesoul@naver.com (D.-H.S.); nohsinwoo@naver.com (S.-W.N.); 2Department of Animal Resources Technology, Gyeongnam National University of Science and Technology, Jinju 52725, Korea; ham8925@gmail.com; 3Department of Animal Science, Research Institute for Functional Halal Animal Products, Chonnam National University, Gwangju 61186, Korea; kbchin@chonnam.ac.kr

**Keywords:** chicken breast, ionic strength, low salt, pre-rigor salting, potassium chloride

## Abstract

The objective of this study is to evaluate the effects of NaCl and KCl salting on technological properties of pre- and post-rigor chicken breasts at various ionic strengths. The following factorial arrangement was used: 2 salt types (NaCl and KCl) × 2 rigor statuses (pre- and post-rigor) × 4 ionic strengths (0.086, 0.171, 0.257, and 0.342). Hot-boned and ground chicken breasts were salted within 30 min postmortem after slaughter (pre-rigor salting) or 24 h postmortem (post-rigor salting) with varying concentrations of NaCl (0.50%, 1.00%, 1.50%, and 2.00%) or KCl (0.64%, 1.28%, 1.91%, and 2.55%) corresponding to the four ionic strengths. KCl caused higher pH value in salted chicken breasts than NaCl (*p* < 0.05). However, KCl decreased total and myofibrillar protein solubilities in post-rigor salted chicken breasts compared to NaCl (*p* < 0.05), but those were similar to pre-rigor chicken breasts, regardless of the salt type (*p* > 0.05). Different salt types had no significant impact on cooking loss and textural properties. This study shows that NaCl and KCl had similar effects on technological properties at the same ionic strength (within 0.342), but the use of KCl may have the possibility to decrease protein solubility, depending on rigor status of raw meat at the different salting time.

## 1. Introduction

In meat processing, sodium chloride (NaCl) is a common ingredient to improve technological properties, such as water-holding capacity, emulsifying capacity, and textural characteristics as well as organoleptic properties and microbial safety [1]. However, it has been well-known that modern consumers in developed countries have negative perceptions regarding excessive sodium intake from processed meat products [2]. Thus, the meat processing industry has been trying to reduce sodium content in processed meat products, with minimizing NaCl addition.

As a practical strategy, the replacement of NaCl with other chloride salts, such as potassium chloride (KCl), calcium chloride (CaCl_2_), and magnesium chloride (MgCl_2_), has been considered [2]. In particular, a 50:50 NaCl:KCl mixture has been proposed as a practical substitute to reduce sodium level in reduced-/low-sodium meat products [1]. In this regard, it has been generally recognized that the 50% NaCl replacement by KCl has no adverse impacts on quality characteristics of the final products [2,3]. However, some previous studies have revealed negative impacts of KCl application on flavor and texture of processed meat products [1,4]. Despite the different ionic strength of NaCl and KCl at the same molar concentration, moreover, little is known about technological properties of meat salted with NaCl and KCl at equivalent ionic strengths.

In general, biochemical and physiochemical conditions of raw meat are important factors affecting salting effects on quality attributes of processed meat products, such as water-holding capacity and texture [5]. In this regard, it has been well documented that pre-rigor salting to maintain high pH and ATP content in pre-rigor skeletal muscles has been known to improve physicochemical properties of meat products, such as cooking yield, emulsion stability, and texture [6]. For these reasons, the practical use of pre-rigor muscle is still considered for the improvement in quality attributes of processed meat products [7,8]. Moreover, it has been proposed that the use of pre-rigor muscle as a raw material is a promising way to produce desirable quality attributes of low-sodium meat products [2].

According to the practical use of pre-rigor salted meat, a previous study has found that the addition of least 2% NaCl should be required to guarantee the positive impacts of pre-rigor salting on water-holding capacity and protein solubility [5]. However, a recent study reported that pre-rigor salting with KCl was less effective compared to pre-rigor salting with NaCl at the same percentage concentration (2%) [9]. Taken together, although it would be expected that pre-rigor salted muscle, particularly with KCl, may be an excellent source for manufacturing low-sodium meat products, there has been little to no information on the undesirable impacts of KCl salting on pre-rigor muscle. In this regard, it could be hypothesized that salting effects of NaCl and KCl on technological properties may be differently affected by rigor status (pre- vs. post-rigor) at salting time. However, there is no available literature for comparing the salting effects of NaCl and KCl on the technological properties of pre- and post-rigor muscles at different ionic strengths. Therefore, this study was performed to compare the salting effects of NaCl and KCl on technological properties of pre- and post-rigor chicken breasts at various ionic strengths.

## 2. Materials and Methods

### 2.1. Experimental Design 

In this study, two salt types (NaCl and KCl), two rigor statuses (pre- and post-rigor), and four ionic strengths (0.86, 0.171, 0.257, and 0.342) were used in a 2 × 2 × 4 factorial design in three randomized complete blocks.

### 2.2. Sample Preparation 

In total, 60 broilers (Ross 308 genotype, 32 days old, 20 birds per block) were commercially raised and conventionally slaughtered at a local abattoir (M company, Dongducheon, Korea). After electric-stunning, exsanguination, scalding, and plucking (7 ± 1 min after stunning), breast muscles (*M. pectoralis major* and *minor*) were manually deboned from randomly selected chicken carcasses from an automated slaughter line. The average time required for deboning was approximately 20–25 min after slaughter in all selected cases. The left- and right-sides of chicken breast muscles were assigned into pre-rigor and post-rigor groups, respectively. The left-side muscles were immediately ground using a meat grinder equipped with a 6 mm plate. The ground pre-rigor chicken breasts were divided into eight equal portions, and each portion was individually mixed with four different NaCl (0.50%, 1.00%, 1.50%, and 2.00%, *w*/*w*) or KCl (0.64%, 1.28%, 1.91%, and 2.55%, *w*/*w*) concentrations. The salted pre-rigor chicken breast samples were vacuum packaged, stored in a 4 °C refrigerator for 24 h, and used for further analysis; whereas the intact right-side muscles were immediately vacuum-packaged in polyamide-polyethylene (PA/PE) film bags and stored in a 4 °C refrigerator for 24 h. The vacuum-packaged post-rigor muscles were ground at 24 h postmortem and salted as mentioned above. Prior to both pre- and post-rigor salting processes, the temperature, pH, and surface color were measured to confirm muscle characteristics of pre- and post-rigor chicken breasts. The temperature, pH value, and color of pre- and post-rigor chicken breasts were measured at postmortem 30 min and 24 h, respectively. The salted post-rigor chicken breasts were individually vacuum packaged, also stored in a 4 °C refrigerator for 24 h and used for further analysis.

Since NaCl is the most common type of salt used in commercial meat processing, on the basis of NaCl concentrations, each ionic strength was calculated using Equation (1). The ionic strengths corresponding to 0.50%, 1.00%, 1.50%, and 2.00% NaCl were 0.86, 0.171, 0.257, and 0.342, respectively. Based on the ionic strengths, KCl concentrations with identical ionic strength were calculated as 0.64%, 1.28%, 1.91%, and 2.55%, respectively.
(1)$$\mathrm{Ionic}\;\mathrm{strength}\;\left(\mathrm{IS}\right)=\frac{1}{2}\sum_{i=1}^{n}c_{i}z_{i}^{2}$$
where, *c_i_* = the concentration of the *i* ion and *z_i_* = the charge of the *i* ion.

### 2.3. Analysis of Intact Chicken Breasts

#### 2.3.1. Temperature and pH Value

The temperature and pH value of intact chicken breasts were measured three times using a portable electric pH meter (HI99163, Hanna instruments, Woonsocket, RI, USA) equipped with a stainless-steel piercing blade.

#### 2.3.2. Color Measurement

The surface color of intact chicken breasts (bone side) was measured using a colorimeter (CR-400, Minolta, Osaka, Japan) equipped with an 8 mm diameter aperture with 2° standard observer. The color of pre- and post-rigor chicken breasts was measured at postmortem 30 min and 24 h, respectively. According to the manufacturer’s manual, the instrument was calibrated using a calibration tile (CIE L*: +93.01, CIE a*: −0.25, CIE b*: +3.50) under a D_65_ illumination source. CIE L*, a*, and b* values were recorded from six random locations.

### 2.4. Analysis of Salted Chicken Breasts

#### 2.4.1. pH Value

The pH values of salted chicken breast samples were measured three times using a portable electric pH meter (HI99163, Hanna instruments).

#### 2.4.2. Cooking Loss

For the determination of cooking loss, salted chicken breast sample (approximately 30 g) was stuffed into a 50 mL centrifuge tube and heated in a 75 °C water bath (JSIB-22T, JS Research Inc., Gongju, Korea) for 30 min [9]. Percentage weight difference between raw and cooked samples was calculated as follows: cooking loss (%) = ((raw sample weight (g) − cooked sample weight (g))/raw sample weight (g)) × 100.

#### 2.4.3. Protein Solubility

Protein solubilities of total and sarcoplasmic proteins in salted chicken breast samples were determined according to the method described in Warner et al. [10]. To determine total protein solubility, 2 g of sample were homogenized with 20 mL of extraction buffer (1.1 M potassium iodide in 0.1 M potassium phosphate buffer, pH 7.2), using a homogenizer (HG-15A, Daehan Sci., Seoul, Korea) at 12,000 rpm for 2 min, and the samples were stored in a 4 °C refrigerator overnight. The homogenate was centrifuged at 1500× *g* for 20 min (4 °C), and the supernatant was filtered through Whatman No. 1 filter paper. The protein concentration of the filtrate was determined by the Biuret method [11] using standard curve with a bovine serum albumin. Sarcoplasmic protein solubility was measured using a different extraction buffer (0.025 M potassium phosphate buffer, pH 7.2) with the same procedure. Myofibrillar protein solubility was calculated from the difference between total and sarcoplasmic proteins. Total protein content of samples was determined using a nitrogen analyzer (Rapid N Cube, Elementar, Langenselbold, Germany). Protein solubility was expressed as soluble protein content/total protein content × 100.

#### 2.4.4. Sodium Dodecyl Sulfate Poly-Acrylamide Gel Electrophoresis (SDS-PAGE)

SDS-PAGE was used to analyze pattern and intensity of soluble proteins in salted chicken breast sample by the Laemmli method [12], with 12% separating and 4% stacking gels. The extracted total protein fraction was diluted with 1.1 M potassium iodide in 0.1 M potassium phosphate buffer (pH 7.2). The extracted samples were then mixed with four volumes of 5× Laemmli sample buffer (312.5 mM Tris-HCl (pH 6.8), 50% glycerol, 5% SDS, 5% β-mercaptoethanol, 0.05% bromophenol blue; EBA-1052, Elpisbiotech, Daejeon, Korea) and boiled at 100 °C for 5 min. Each sample (60 μg) was loaded onto the gel and electrophoresis was performed at 100 V for 2 h. The loaded gel was stained with Coomassie Brilliant Blue R250 (B7920, Sigma, Saint Louis, MO, USA) and de-stained in a solution of methanol, distilled water, and acetic acid (50:40:10). The separated protein bands were identified by comparing molecular weights with a standard protein marker (pre-stained DokDo-MARK, EBM-1032, Elpisbiotech, Daejeon, Korea).

#### 2.4.5. Emulsion Activity Index (EAI)

EAI values of salted chicken breast samples were evaluated using the method published and modified by Pearce and Kinsella [13] as described by Chan et al. [14]. The extracted total protein fraction was diluted with 1.1 M potassium iodide in 0.1 M potassium phosphate buffer (pH 7.2). Then, 20 mL of total protein fraction (1.5 mg/mL) were homogenized with 6.6 mL of commercial soybean oil (Pure refined soybean oil, Sajo Daerim Corp., Seoul, Korea) at 14,000 rpm for 1 min. Then, 50 μL of the emulsified solution was mixed with 5 mL of 0.1% SDS solution, and the absorbance of the mixture was read in quadruple at 500 nm. EAI was calculated as follows EAI = absorbance at 500 nm × 2.33.

#### 2.4.6. Texture Profile Analysis

Texture profile analysis of the cooked samples was performed using a texture analyzer (CT3, Brookfield Engineering Laboratories, Inc., Middleboro, MA, USA). The cooked samples used for the cooking loss determination were equilibrated to room temperature (25 °C) for 3 h, and a total of eight samples (cylinder shaped, 2.5 cm height and 2 cm diameter) per treatment was taken from the central portion of each gel. A compression cycle test (70% compression of the original sample height) was performed twice with a cylinder probe (diameter = 5 cm). Sample deformation curves were obtained with a 50 kg maximum load cell, and the analysis conditions were as follows: pre-test speed = 1.0 mm/s, post-test speed = 5.0 mm/s, and test speed = 2.0 mm/s. Values for hardness (kg), springiness (ratio), cohesiveness (unitless), gumminess (kg), and chewiness (kg) were determined [15].

### 2.5. Statistical Analysis

The experimental design of this study was a completely randomized block design with three independent batches (*n* = 3). Analysis of variance was performed on all variables measured using the general linear model (GLM) procedure in SPSS 18.0 program (SPSS Inc., Chicago, IL, USA). Between intact pre-rigor chicken breast and intact post-rigor chicken breast, the significance of difference was determined using Student’s *t*-test (*p* < 0.05). For pre- and post-rigor salting treatments, three-way ANOVA was performed, in which salt type, rigor status, and ionic strength were fixed as the main effects, and their interactions were also considered. *t*-Test and Tukey’s method were used to determine the significance of the differences between treatments (*p* < 0.05).

## 3. Results

### 3.1. Temperature, pH Value, and Color of Intact Pre- and Post-Rigor Chicken Breasts

Temperature, pH value, and color characteristics of intact pre- and post-rigor chicken breasts were evaluated to confirm their different physicochemical characteristics as a raw material (Table 1). The average temperature of pre- and post-rigor chicken breasts was 33.33 °C and 5.67 °C, respectively (*p* < 0.001). The pH value of pre-rigor chicken breasts (6.46) was significantly higher than that of post-rigor chicken breasts (5.87). For color characteristics, pre-rigor chicken breasts showed a lower CIE L * value (lightness) than post-rigor chicken breasts (*p* < 0.001). No difference in CIE a * (redness) was observed between pre- and post-rigor chicken breasts (*p* > 0.05). The difference in CIE b * (yellowness) between pre-and post-rigor chicken breasts was small, but significant. Similar results for the lower lightness in pre-rigor chicken breasts were observed by Qiao et al. [16] and Perlo et al. [17], who observed an increase in lightness of chicken breasts with increasing postmortem storage duration. According to Qiao et al. [16], lightness of chicken breasts at 0 h and 24 h after postmortem was negatively correlated with pH value (*r* = −0.9632 and −0.9610, respectively; *p* = 0.0001). Since high pH value of meat generally contributes to an improvement in water-holding capacity (WHC), the lower lightness of pre-rigor chicken breasts could be attributed to their excellent water-holding capacity with high pH value, thereby reducing light-scattering on the meat surface [18].

### 3.2. Technological Properties of Salted Chicken Breasts

The significance of *p*-value from comparing the main effects (salt type, rigor status, and ionic strength) and their two- and three-way interactions are shown in Table 2. No three-way interactions among the main effects were found for any of the measured variables. Results for the single main effects on technological properties are presented in Table 3, and their two-way interaction effects presenting statistically significant difference are shown in Figure 1, Figure 2, Figure 3 and Figure 4. 

#### 3.2.1. pH Value

The pH value of salted chicken breasts was significantly affected by salt type, rigor status, and rigor status × ionic strength interaction effects (Table 2). As shown in Table 3, KCl led to higher pH value of salted chicken breasts compared to NaCl (5.93 vs. 5.84; *p* < 0.05). According to previous studies, Keeton [19] reported that pork hams salted with KCl showed a higher pH value (6.17) than those salted with NaCl (6.05). Hand et al. [20] noted that partial or complete replacement of NaCl with KCl, at equivalent ionic strengths, increased the pH value in cured pork ham. Although the exact mechanism behind increased pH value in meat salted with KCl has not been clearly understood, Aliño et al. [21] suggested that such a phenomenon might be related to differences in binding abilities of sodium and potassium ions to muscle protein and/or in cell membrane permeability.

As expected, the pH value in pre-rigor salted chicken breasts were significantly higher than those of post-rigor salted chicken breasts (5.98 vs. 5.80). In addition, an interaction between rigor status and ionic strength on pH value was found, indicating that the pH value of pre- and post-rigor salted chicken breasts was differently affected by ionic strength (Figure 1a). Many previous studies reported that pre-rigor salting could minimize postmortem pH value decline due to the inactivation of enzymes related to postmortem anaerobic glycolysis, such as phosphorylase and phosphofructokinase [5,22]. According to Kim et al. [5], in pre-rigor chicken breast muscle, high ionic strengths corresponding to 1.5–2.0% NaCl concentrations could be required to maintain high pH value of pre-rigor muscle. In pre-rigor salted chicken breasts, thus, an increase in ionic strength could inhibit postmortem glycolysis, which could in turn contribute to the maintenance of high pH value. Conversely, in post-rigor muscle where postmortem glycolysis is completed, decreased pH value with increasing ionic strength was found in this study. This result could be associated with increased hydrophilic group exposure on thick filament shafts, because they are depolymerized at high ionic strength conditions [23].

#### 3.2.2. Cooking Loss

The cooking loss of salted chicken breasts was affected by rigor status (*p* = 0.015), ionic strength (*p* < 0.001), and rigor status × ionic strength interaction (*p* = 0.017), but not by salt type (*p* > 0.05) (Table 2). These results showing the rigor status × ionic strength interaction effect on cooking loss are shown in Figure 1b. No significant differences in cooking loss between pre- and post-rigor salted chicken breasts were found at ionic strengths of 0.086 and 0.171. However, at both ionic strengths 0.257 and 0.342, cooking losses of pre-rigor salted chicken breasts were lower than those of post-rigor salted chicken breasts, but it was not significant. Previously, it has been well documented that pre-rigor salted meat had better water-holding capacity compared to post-rigor salted muscle, because pre-rigor muscle had a higher pH value and protein solubility/extractability than post-rigor muscle [5,24]. According to Bernthal et al. [25], the addition of at least 2% NaCl would be required to guarantee the positive impact of pre-rigor salting on water-holding capacity of ground beef. Similarly, Kim et al. [5] found that the addition of <2% NaCl had little to no positive impact on cooking loss of ground pre-rigor salted chicken breasts, which was also confirmed in this current study. Thus, our results are consistent with previous observations, and further suggest that salt type (NaCl vs. KCl) may have no impact on cooking loss of salted chicken breasts, irrespective of their rigor status.

#### 3.2.3. Protein Solubilities

Total and myofibrillar protein solubilities of salted chicken breasts were affected by all main effects and salt type × rigor status interaction effect (Table 2). Pre-rigor salting and high ionic strength led to an increase in total and myofibrillar protein solubilities (*p* < 0.05; Table 3). However, sarcoplasmic protein solubility from salted chicken breasts was not affected by salt type, or their interaction with other independent variables (*p* > 0.05). KCl caused slightly lower total and myofibrillar protein solubilities of salted chicken breasts compared to NaCl (*p* < 0.05). Given the salt type × rigor status interaction effect (Figure 2a,b), however, different salt types had no impact on total and myofibrillar protein solubilities in pre-rigor salted chicken breasts. On the other hand, in post-rigor salted chicken breasts, KCl resulted in slightly lower total and myofibrillar protein solubilities compared to NaCl (*p* < 0.05). Similarly, Wu et al. [26] found that KCl was less effective in solubilizing myofibrillar proteins, extracted from porcine loin at 12 h postmortem, than NaCl at the same salt concentration (0.6 M at pH 5.5), and suggested that it was likely due to differences in the hydration degree between sodium and potassium ions. Previously, Salis and Ninham [27] reported that the binding ability of cations to negatively charged proteins was as follows: Li^+^ > Na^+^ > K^+^. Thus, the decreased total and myofibrillar protein solubilities in post-rigor salting with KCl was likely due to the relatively weak interaction of potassium ion with muscle proteins. In general, pre-rigor muscle has a swollen structure with weakened myosin-actin cross-bridges in myofibrils [28], in which the difference in binding ability between cations might be relatively less impactful on protein solubility as compared to post-rigor muscle. Consequently, our results confirmed that rigor status of raw meat greatly affects myofibrillar protein solubility. In addition, the type of salt (NaCl and KCl) at the same ionic strength could differently influence total and myofibrillar protein solubilities of pre-rigor chicken breasts.

#### 3.2.4. SDS-PAGE

Proteins in myofibrils isolated from salted chicken breasts were observed by SDS-PAGE system (Figure 3). Myosin heavy chains (200 kDa), α-actinin (100 kDa), desmin (53 kDa), G-actin (42 kDa), and myosin light chains (15–20 kDa) were observed similarly in all treatments [29]. Thus, it seems that the patterns of extractable myofibrillar proteins from salted chicken breasts might not be affected by salt type, rigor status, or ionic strength. As a similar result, Munasinghe and Sakai [30] reported that the SDS-PAGE patterns of extractable proteins from pork muscle were unaffected by salt type (NaCl, KCl vs. LiCl) and pH value difference within physiological range (pH 6.0–8.0). Moreover, Xiong et al. [31] reported no remarkable degradation of muscle proteins in post-rigor chicken breast.

#### 3.2.5. Emulsion Activity Index (EAI)

No significant differences in EAI of total protein fraction extracted from salted chicken breasts were found at the same protein concentration (Table 2 and Table 3). These results may indicate that extracted total protein fraction may have comparable emulsifying capacity at the same protein concentration, regardless of salt type, rigor status, or ionic strength.

#### 3.2.6. Textural Profile Analysis

All textural parameters, except for springiness, were significantly affected by rigor status or ionic strength (Table 2). However, salt type had no impacts on all measured textural parameters, regardless of rigor status and ionic strength (*p* > 0.05; Table 3). Although significant salt type × rigor status interaction effects were found on cohesiveness and chewiness (Figure 4a,b), the cohesiveness and chewiness of chicken breasts salted with NaCl and KCl were similar within each rigor status, respectively (*p* > 0.05). Pre-rigor salted chicken breasts exhibited increased hardness, cohesiveness, gumminess, and chewiness than post-rigor salted chicken breasts (*p* < 0.05; Table 3). Previously, Kim et al. [5] reported that the hardness of pre-rigor salted chicken breasts was increased with increasing NaCl levels (0% to 2%). In this study, an increase in ionic strength led to increases in both primary (hardness, springiness, and cohesiveness), and secondary (gumminess and chewiness) textural parameters (*p* < 0.05). In particular, pre-rigor chicken breast salted at ionic strength of 0.342 showed the highest cohesiveness and chewiness (*p* < 0.05; Figure 4c,d), indicating greater salting impacts on the textural parameters of pre-rigor muscle as compared to post-rigor muscle at the high ionic strength.

## 4. Conclusions

This current study confirms the positive impacts of pre-rigor salting on technological properties of salted chicken breasts. Moreover, pre-rigor salting with ionic strengths of 0.257, corresponding to 1.5% NaCl or 1.91% KCl, could be required to improve the cooking loss and textural properties. In terms of salt type effect (NaCl vs. KCl), KCl was less effective in solubilizing total and myofibrillar proteins in salted post-rigor chicken breasts. However, the differences in total and myofibrillar protein solubilities between NaCl and KCl had no impacts on cooking loss and textural properties. Thus, it could be concluded that NaCl and KCl could have comparable salting effects on technological properties if used at the same ionic strength. However, their salting effects on protein solubility/extractability of muscle proteins might be differently affected by rigor status of raw meat. Considering the complex nature of meat product manufacturing processes, it could be suggested that partial and/or complete replacement of NaCl with KCl under certain processing conditions may have the potential possibility of deteriorating technological properties of final products, in relation to a decrease in protein solubility.

## Figures and Tables

**Figure 1 foods-09-00721-f001:**
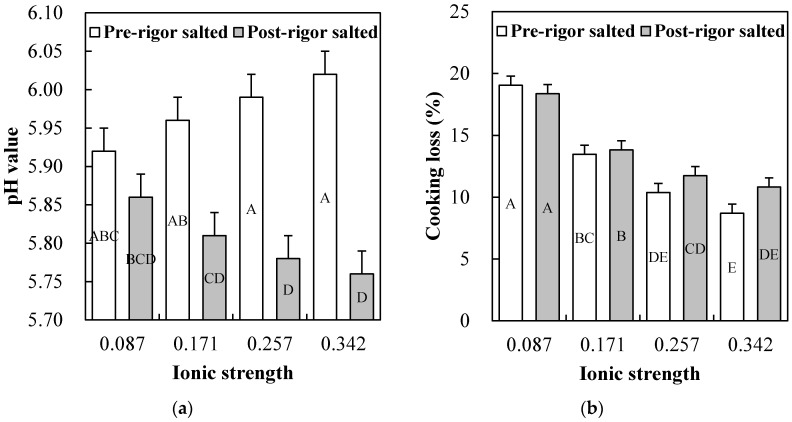
Rigor status (pre- vs. post-rigor salted) × ionic strength (0.087, 0.171, 0.257 vs. 0.34) interaction effects on pH value (**a**) and cooking loss (**b**) of salted chicken breasts. Error bars represent standard error of the means. A–E: means with the same letter are not significantly different (*p* ≥ 0.05).

**Figure 2 foods-09-00721-f002:**
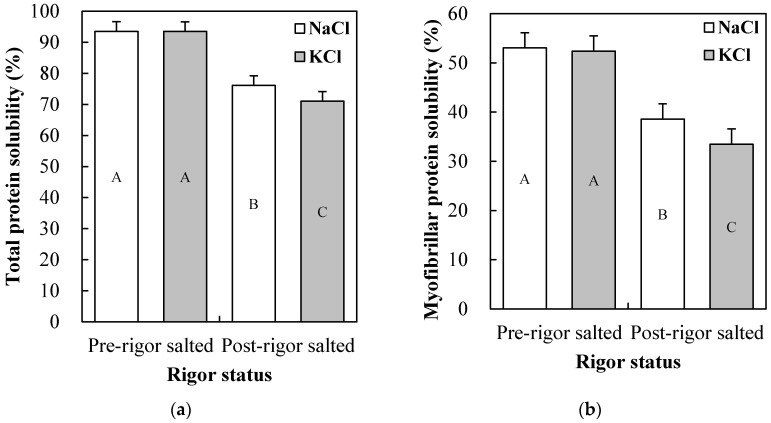
Salt type (NaCl vs. KCl) × rigor status (pre- vs. post-rigor salted) interaction effects on total (**a**) and myofibrillar protein solubility (**b**) of salted chicken breasts. Error bars represent standard error of the means. A–C: means with the same letter are not significantly different (*p* ≥ 0.05).

**Figure 3 foods-09-00721-f003:**
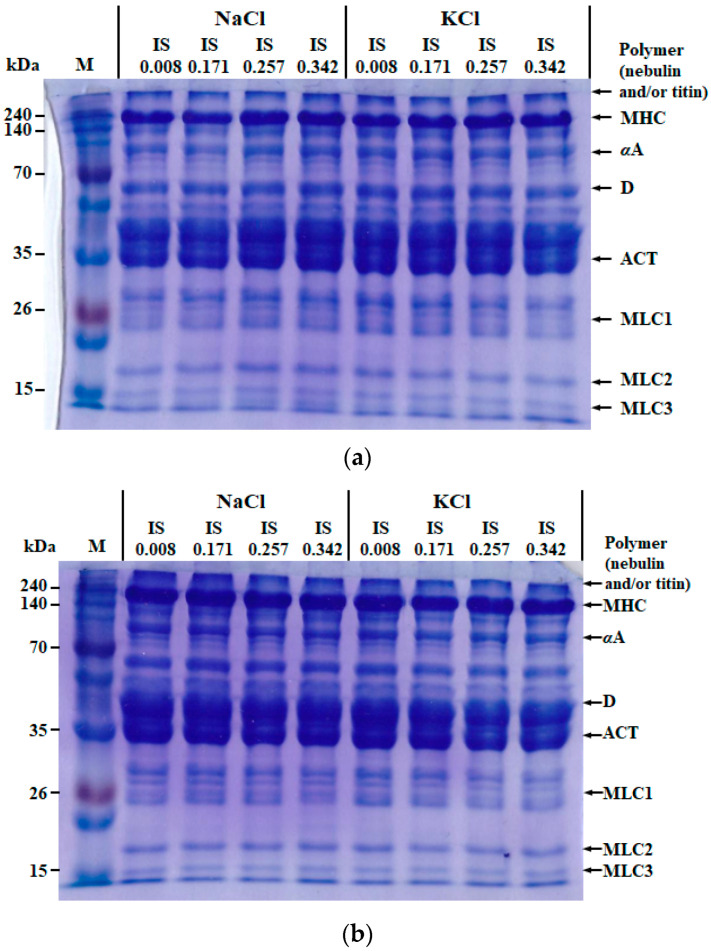
Representative sodium dodecyl sulfate poly-acrylamide gel electrophoresis (SDS-PAGE) photos of colloid Coomassie blue-stained 12.5% polyacrylamide gels of pre- (**a**) and post-rigor (**b**) salted chicken breast with NaCl or KCl at different ionic strengths. IS, ionic strength; M, standard protein marker; polymer, probably nebulin and/or titin; MHC, myosin heavy chain; αA, α-actinin; D, desmin; ACT, actin; MLC, myosin light chain.

**Figure 4 foods-09-00721-f004:**
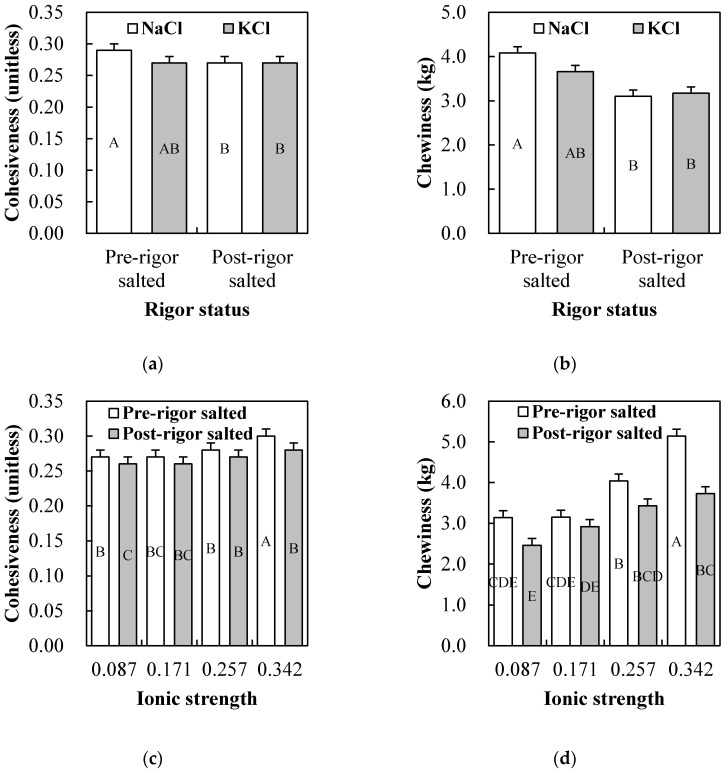
Salt type (NaCl vs. KCl) × rigor status (pre- vs. post-rigor salted) or rigor status (pre- vs. post-rigor salted) × ionic strength (0.087, 0.171, 0.257 vs. 0.34) interaction effects on cohesiveness (**a**,**c**) and chewiness (**b**,**d**) of salted chicken breasts. Error bars represent standard error of the means. A–E: means with the same letter are not significantly different (*p* > 0.05).

**Table 1 foods-09-00721-t001:** Temperature, pH value, and color characteristics of intact pre- and post-rigor chicken breasts.

Traits		Pre-Rigor Chicken Breasts	Post-Rigor Chicken Breasts	SEM ^(1)^	Significance of *t*-Test ^(2)^
Temperature (°C)		33.33	5.67	6.51	***
pH value		6.46	5.87	0.13	***
Color	CIE L * (lightness)	46.87	53.22	1.44	***
	CIE a * (redness)	2.67	2.68	0.06	NS
	CIE b * (yellowness)	4.94	3.79	0.29	***

^(1)^ SEM: standard error of the means; ^(2)^ NS: non-significance (*p* ≥ 0.05); *** *p* < 0.001.

**Table 2 foods-09-00721-t002:** Significance of *p*-value of main effects and their interactions on measured variables of salted chicken breasts.

Measured Variables	Independent Variables (Main Effects)			Two- and Three-Way Interactions			
	Salt Type	Rigor Status	Ionic Strength	S ^(1)^ × R ^(2)^	R × I ^(3)^	S × I	S × R × I
pH value	<0.001	<0.001	NS ^(5)^	NS	<0.001	NS	NS
Cooking loss (%)	NS	0.015	<0.001	NS	0.017	NS	NS
Protein solubility (%)							
Total protein	0.003	<0.001	<0.001	0.006	NS	NS	NS
Sarcoplasmic protein	NS	<0.001	NS	NS	0.013	NS	NS
Myofibrillar protein	0.003	<0.001	<0.001	0.027	NS	NS	NS
EAI ^(4)^	NS	NS	NS	NS	NS	NS	NS
Textural properties							
Hardness (kg)	NS	<0.001	<0.001	NS	NS	NS	NS
Springiness (unitless)	NS	NS	<0.001	NS	NS	NS	NS
Cohesiveness (ratio)	NS	<0.001	<0.001	0.021	0.032	NS	NS
Gumminess (kg)	NS	<0.001	<0.001	NS	NS	NS	NS
Chewiness (kg)	NS	<0.001	<0.001	0.036	0.005	NS	NS

^(1)^ S: salt type; ^(2)^ R: rigor status; ^(3)^ I: ionic strength; ^(4)^ EAI: emulsion activity index; ^(5)^ NS: non-significance (*p* ≥ 0.05).

**Table 3 foods-09-00721-t003:** Effects of salt type, rigor status, and ionic strength on technological properties of salted chicken breasts.

Effects	pH Value	Cooking Loss (%)	Protein Solubility (%)			EAI ^(1)^	Textural Properties				
			Total Protein	Sarcoplasmic Protein	Myofibrillar Protein		Hardness (kg)	Springiness (Ratio)	Cohesiveness (Unitless)	Gumminess (kg)	Chewiness (kg)
Salt type effect											
NaCl	5.84b	13.16	84.84a	39.16	45.80a	1.54	18.96	0.68	0.28	5.24	3.59
KCl	5.93a	13.42	82.29b	39.35	42.94b	1.49	18.73	0.67	0.27	5.04	3.41
SEM ^(2)^	0.03	1.30	0.82	0.31	0.80	0.07	0.57	0.02	0.01	0.23	0.24
Rigor status effect											
Pre-rigor	5.98a	12.90b	93.53a	40.81a	52.72a	1.54	20.30a	0.68	0.28a	5.67a	3.87a
Post-rigor	5.80b	13.68a	73.60b	37.70b	36.02b	1.46	17.39b	0.68	0.27b	4.61b	3.13b
SEM	0.04	0.31	4.49	0.77	3.79	0.10	0.71	0.01	0.01	0.26	0.19
Ionic strength effect											
0.087	5.89	18.70a	79.48b	39.72	39.98b	1.36	17.24b	0.62c	0.26b	4.55b	2.80c
0.171	5.88	13.64b	82.07b	39.10	42.97b	1.44	17.69b	0.66b	0.27b	4.67b	3.03c
0.257	5.89	11.06c	85.83a	39.34	46.51a	1.53	19.43ab	0.71a	0.27ab	5.31ab	3.74b
0.342	5.89	9.76d	86.88a	38.87	48.03a	1.70	21.02a	0.73a	0.29a	6.04a	4.44a
SEM	0.01	1.04	0.98	0.24	1.04	0.06	0.52	0.01	0.01	0.20	0.20

a,b,c: sharing the same letters in a column within each main effect are not significantly different (*p* ≥ 0.05); ^(1)^ EAI: emulsion activity index; ^(2)^ SEM: standard error of the means.

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
