# Peer review of "Evaluation of NaCl and KCl Salting Effects on Technological Properties of Pre- and Post-Rigor Chicken Breasts at Various Ionic Strengths"

_foods, 2020, doi:10.3390/foods9060721_

Round 1

Reviewer 1 Report

Review of the manuscript titled:

Evaluation of NaCl and KCl Salting Effects on Technological Properties of Pre- and Post-Rigor Chicken Breasts at Various Ionic Strengths

General comments:

The manuscript should be in the interest on meat/ poultry manufacturers. The undertaken study is in takes about the popular requirement of modern consumers, of reduced content of salt in processed meat. The arrangement of the experiment is interesting, and proper. The methods were properly selected, however the description should be improved as in its current form it’s confusing. The results and discussion sections are presented transparently and  are factually correct. English requires improvement, I am not a native, but I can see a lot of stylistic errors. The manuscript can be accepted for publication after minor revision.

Detailed comments:

Lines 76-95.

The description of procedures implemented on pre-rigor and post-rigor chicken samples should be presented chronologically. First, all the procedures on pre-rigor samples should be described. Then, all procedures on post-rigor samples. Finally, the ionic strength calculation.

Lines 98-100

The time of measurements should be given here – it is presented under one of the tables, but it’s not a proper place to describe the methodology. Also, it is not clear if the pH was measured on both, pre- and post-rigor samples, as the only information given here, is that these were intact samples?? Of course the reader can figure it out from the results description and the tables, but it should be clearly indicated in the Materials and methods section.

Line 99

As the pH meter temperature compensated, was it equipped in an electrode (give the type, and company name).

Line 102-106

What time post-mortem was the colour measured? Which observer angle was used (2áµ’ or 10áµ’)? The colour was measured pre-rigor and post-rigor? If yes, please describe it in this section.

Table 1. As mentioned before, the time of measurements should not be given under the table, but in the description of methodology.

Table 2. EAI, S, R, I – you should make a use of these abbreviations. I suggest to use these abbreviations in the table’s main text, and explain them under the table.

Table 3. EAI – please define under the table

Author Response

Response to Reviewer 1 Comments

The manuscript should be in the interest on meat/ poultry manufacturers. The undertaken study is in takes about the popular requirement of modern consumers, of reduced content of salt in processed meat. The arrangement of the experiment is interesting, and proper. The methods were properly selected, however the description should be improved as in its current form it’s confusing. The results and discussion sections are presented transparently and are factually correct. English requires improvement, I am not a native, but I can see a lot of stylistic errors. The manuscript can be accepted for publication after minor revision.

Thank you for all your comments and consideration on this review process. Considering your valuable comments, we has appropriately revised all points for improving the quality of this manuscript. Please find the revised parts with tracking mode of Word program. Thanks again.

Detailed comments:

Point 1(L76-95): The description of procedures implemented on pre-rigor and post-rigor chicken samples should be presented chronologically. First, all the procedures on pre-rigor samples should be described. Then, all procedures on post-rigor samples. Finally, the ionic strength calculation.

Response 1: According to your suggestion, we chronologically revised the sentences for description of sample preparation. Please see line 76-93 in the revised version of the manuscript.

Point 2(L98-100): The time of measurements should be given here – it is presented under one of the tables, but it’s not a proper place to describe the methodology. Also, it is not clear if the pH was measured on both, pre- and post-rigor samples, as the only information given here, is that these were intact samples?? Of course the reader can figure it out from the results description and the tables, but it should be clearly indicated in the Materials and methods section.

Response 2: We revised the parts for clearly showing all experimental conditions at pH measurement. Please see line 76-93 in the revised version of the manuscript.

Point 3(L99): As the pH meter temperature compensated, was it equipped in an electrode (give the type, and company name).

Response 3: We provided the information on the pH meter originally equipped with an electrode (model, company, and country). Please see line 105-107 in the revised version of the manuscript.

Point 4(L102-106): What time post-mortem was the colour measured? Which observer angle was used (2áµ’ or 10áµ’)? The colour was measured pre-rigor and post-rigor? If yes, please describe it in this section.

Response 4: We described the time at color measurement and the information (illuminant, aperture, and observer) on colorimeter used. Please see line 109-114 in the revised version of the manuscript.

Point 5(Table 1): As mentioned before, the time of measurements should not be given under the table, but in the description of methodology.

Response 5: According to your suggestion, we deleted methodological information described in footnote of Table 1, but described the information in M&M as mentioned in  Response 2.

Point 6(Table 2): EAI, S, R, I – you should make a use of these abbreviations. I suggest to use these abbreviations in the table’s main text, and explain them under the table.

Response 6: According to your suggestion, we revised Table 2 and its footnote. Please see Table 2 in the revised version of the manuscript.

Point 7(Table 3): EAI – please define under the table.

Response 7: According to your suggestion, we described the definition of EAI in the footnote in Table 3. Please see Table 3 in the revised version of the manuscript.

ttachment.

Reviewer 2 Report

Evaluation of NaCl and KCl salting effects on technological properties of pre and post rigor chicken breasts at various ionic strengths

The paper deals with effect of different salt with different concentration, pre and post rigor on the pH, texture, protein solubility. The paper about Evaluation of NaCl and KCl salting effects on technological properties of pre and post rigor chicken breasts at various ionic strengths is of general interest; however, I would like to draw the attention on the following points:

The novelty of the work (questionable, e.g., how this work is different from previous works, line 346 and 356, e.g., number 5 and 9 in the reference list from the same Authors), contribution of the work is not clear, and should be clear and also justified with state of art…. The hypothesis of using the NaCl need to be justified, why? For the mechanism needs to be described, why the KCl is better and different from Nacl? How about the nutritional values and other functionality, is there any other studies reported? Why the KCl was not used until now? But This is not need to be clearly justified with sufficient clarity and literatures. The material and method needs to clear, and the statistical method is not clear. There are a lot of language mistakes and unclearity, should be rewritten to resolve these issues. Moreover, the following specific comments:

  • Abstract section: “ NaCl (0.50, 1.00, 1.50, and 2.00%) or KCl (0.64, 1.28, 1.91, and 2.55%) corresponding to the four ionic strengths.” Why the concentration of two salt set at different values instead the same level?
    • “This study shows that NaCl and KCl had similar effects on technological properties at the same ionic strength (within 0.342), but the use of KCl may have the possibility to decrease protein solubility, depending”
  • Page 1, Line 33 , it is vague, what do you mean by the perception? is that perception or ….?
  • What are the adverse effect of other salt such KCL, and other compared to other salts? What about the nutritional value or functionality compared to other Nacl? Why also little knowledge about the technological of the salts?
  • Section 2 (material and methods)
    • Experimental “Arrangement”, use another word to make clear
    • Equation is loose, it should be written in more clearly and with better structure (e.g. what is the units, equation from the text, avoid mix of equation and text, use equation to write equation …etc)
    • Section 2.3.1, and 2.3.2, are not clear? How?
    • What is difference between 2.3.1 and 2.3.1. regarding the ph measurement?
    • Section 2.4.2 is not clear
  • The statistical analysis is not clear
    • Table 1, Table 2 are not clear ( the statistical results are not clearly presented and discussing the results section)
  • Line 216, can the pH inactivates the enzymes? Or inhibit? What about effect of lower ph on the enzyme activities?
  • Table 3 is not discussed well.

Line 241-243, what is the hypothesis of this paper? If other studies already proved the salt type, does not have

Author Response

Response to Reviewer 2 Comments

The paper deals with effect of different salt with different concentration, pre and post rigor on the pH, texture, protein solubility. The paper about Evaluation of NaCl and KCl salting effects on technological properties of pre and post rigor chicken breasts at various ionic strengths is of general interest; however, I would like to draw the attention on the following points:

The novelty of the work (questionable, e.g., how this work is different from previous works, line 346 and 356, e.g., number 5 and 9 in the reference list from the same Authors), contribution of the work is not clear, and should be clear and also justified with state of art…. The hypothesis of using the NaCl need to be justified, why? For the mechanism needs to be described, why the KCl is better and different from Nacl? How about the nutritional values and other functionality, is there any other studies reported? Why the KCl was not used until now? But This is not need to be clearly justified with sufficient clarity and literatures. The material and method needs to clear, and the statistical method is not clear. There are a lot of language mistakes and unclearity, should be rewritten to resolve these issues.

Thank you for all your comments and consideration on this review process. Considering your valuable comments, we has appropriately revised all points for improving the quality of this manuscript. Please find the revised parts with tracking mode of Word program. In addition, we would like to address that KCl has been commercially used for replacing the same percentage of NaCl in low-salt meat product, despite their different ionic strengths. In this regard, this study was conducted to compare salting effect of NaCl and KCl in pre- and post-rigor muscles. To our knowledge, there has been little to no information their salting effects at the same ionic strength, and we belive that this point may guarantee the novelty of this study. Regarding language problems, we have done proofreading of this manuscript by a native speaker. Thanks again.

Specific comments:

Point 1(abstract): “ NaCl (0.50, 1.00, 1.50, and 2.00%) or KCl (0.64, 1.28, 1.91, and 2.55%) corresponding to the four ionic strengths.” Why the concentration of two salt set at different values instead the same level?

Response 1: As mentioned above, the processing characteristics of raw meat is differently affected by ionic strengths produced with different salt concentration. While some previous studies have evaluated the different salting effect of NaCl and KCl at the same percentage concentration, there have been little to no literatures regarding the salting effect of NaCl and KCl at the same ionic strength. Thus, we believe that this could be an interesting point and reasonable research objectives.

Point 2(abstract): This study shows that NaCl and KCl had similar effects on technological properties at the same ionic strength (within 0.342), but the use of KCl may have the possibility to decrease protein solubility, depending”

Response 2: It is hard to understand where it is pointed out, but we double-checked the information and grammar erros.

Point 3(P1L33): it is vague, what do you mean by the perception? is that perception or ….?

Response 3: This sentence means that modern consumers have been trying to avoid excessive sodium intake, which could cause negative perception of the intake of processed meat products with high salt content.

Point 4(L102-106): What are the adverse effect of other salt such KCL, and other compared to other salts? What about the nutritional value or functionality compared to other Nacl? Why also little knowledge about the technological of the salts?

Response 4: This study has been not conducted to compare the nutritional and functional properties of KCl, in general, since the addition of salt in meat products is for processing/technical purposes rather than nutritional and functional benefits. Please reconsider the major objective of this study again.

Point 5(M&M): Experimental “Arrangement”, use another word to make clear

Response 5: We revised “arrangement” to “design” for better understanding. Please see line 173-178 in the revised version of the manuscript.

Point 6(M&M): Equation is loose, it should be written in more clearly and with better structure (e.g. what is the units, equation from the text, avoid mix of equation and text, use equation to write equation …etc)

Response 6: We described the equation for calculating ionic strength with all information. Please see line 96-97 in the revised version of the manuscript.

Point 7(M&M): Section 2.3.1, and 2.3.2, are not clear? How?

Response 7: We revised section 2.3.1. and 2.3.2 for providing more information on temperature/pH/color measurements. Please see line 104-114 in the revised version of the manuscript.

Point 8(M&M): What is difference between 2.3.1 and 2.3.1. regarding the ph measurement?

Response 8: Section 2.3.1 was structured to explain the methodological description on the pH measurement of intact muscle, whereas section 2.4.1 was for the pH measurement of ground and salted chicken breasts.

Point 9(M&M): Section 2.4.2 is not clear

Response 9: The procedure of cooking loss is one of the common methods to evaluate the water-holding capacity during thermal treatment. Since the methodological procedure is quite general, it is hard to understand what the unclear point is here. If it is still insufficient, please mention it again in more detail.

Point 10(M&M): The statistical analysis is not clear

Response 10: We rephrased the statistical analysis section for readers’ better understanding. Please see line 173-180 in the revised version of the manuscript.

Point 11(M&M): Table 1, Table 2 are not clear ( the statistical results are not clearly presented and discussing the results section)

Response 11: We partially revised Table 1 and 2 for better understanding. Please see Table 1 and 2 in the revised version of the manuscript.

Point 12(L216): can the pH inactivates the enzymes? Or inhibit? What about effect of lower ph on the enzyme activities?

Response 12: Pre-rigor salting could inactivate the enzyme related to glycolysis under high ionic strength. Regarding this point, we cited relevant references (reference 5 and 22).

Point 13(Table 3): Table 3 is not discussed well.

Response 13: We double-checked all results described in results and discussion section. If it is still insufficient, please mention it again in more detail.

Point 14(L241-243): what is the hypothesis of this paper? If other studies already proved the salt type, does not have

Response 14: The hypothesis of this study is that the salting effects of KCl and NaCl on technological properties of chicken breasts may be different at the same ionic strength, and their salting effect would be affected by rigor status of raw meat.

Round 2

Reviewer 2 Report

Most of critical questions raised (e.g., originality or novelty work), are not addressed with justification or strong argument, what is significant scientific contributions, just “commercial”, that is not convincing”. No significant change to manuscript compared to the original submitted manuscript. It should be thoroughly look into the questions raised.

Author Response

Response to Reviewer 2

The paper deals with effect of different salt with different concentration, pre and post rigor on the pH, texture, protein solubility. The paper about Evaluation of NaCl and KCl salting effects on technological properties of pre and post rigor chicken breasts at various ionic strengths is of general interest; however, I would like to draw the attention on the following points:

The novelty of the work (questionable, e.g., how this work is different from previous works, line 346 and 356, e.g., number 5 and 9 in the reference list from the same Authors), contribution of the work is not clear, and should be clear and also justified with state of art…. The hypothesis of using the NaCl need to be justified, why? For the mechanism needs to be described, why the KCl is better and different from Nacl? How about the nutritional values and other functionality, is there any other studies reported? Why the KCl was not used until now? But This is not need to be clearly justified with sufficient clarity and literatures. The material and method needs to clear, and the statistical method is not clear. There are a lot of language mistakes and unclearity, should be rewritten to resolve these issues.

Thank you for all your comments and consideration on this review process. Considering your valuable comments, we have appropriately revised all points for improving the quality of this manuscript. Please find the revised parts with tracking mode of Word program. In addition, we would like to address that KCl has been commercially used for replacing the same percentage of NaCl in low-salt meat product, despite their different ionic strengths. In this regard, this study was conducted to compare salting effect of NaCl and KCl in pre- and post-rigor muscles. To our knowledge, there has been little to no information their salting effects at the same ionic strength, and we belive that this point may guarantee the novelty of this study. Regarding language problems, we have done proofreading of this manuscript by a native speaker. Thanks again.

Specific comments:

Point 1(abstract): “ NaCl (0.50, 1.00, 1.50, and 2.00%) or KCl (0.64, 1.28, 1.91, and 2.55%) corresponding to the four ionic strengths.” Why the concentration of two salt set at different values instead the same level?

Response 1: As mentioned above, the processing characteristics of raw meat is differently affected by ionic strengths produced with different salt concentration. While some previous studies have evaluated the different salting effect of NaCl and KCl at the same percentage concentration, there have been little to no literatures regarding the salting effect of NaCl and KCl at the same ionic strength. Thus, we believe that this could be an interesting point and reasonable research objectives.

Point 2(abstract): This study shows that NaCl and KCl had similar effects on technological properties at the same ionic strength (within 0.342), but the use of KCl may have the possibility to decrease protein solubility, depending”

Response 2: It is hard to understand where it is pointed out, but we double-checked the information and grammar erros.

Point 3(P1L33): it is vague, what do you mean by the perception? is that perception or ….?

Response 3: This sentence means that modern consumers have been trying to avoid excessive sodium intake, which could cause negative perception of the intake of processed meat products with high salt content.

Point 4(L102-106): What are the adverse effect of other salt such KCL, and other compared to other salts? What about the nutritional value or functionality compared to other Nacl? Why also little knowledge about the technological of the salts?

Response 4: This study has been not conducted to compare the nutritional and functional properties of KCl, in general, since the addition of salt in meat products is for processing/technical purposes rather than nutritional and functional benefits. Please reconsider the major objective of this study again.

Point 5(M&M): Experimental “Arrangement”, use another word to make clear

Response 5: We revised “arrangement” to “design” for better understanding. Please see line 173-178 in the revised version of the manuscript.

Point 6(M&M): Equation is loose, it should be written in more clearly and with better structure (e.g. what is the units, equation from the text, avoid mix of equation and text, use equation to write equation …etc)

Response 6: We described the equation for calculating ionic strength with all information. Please see line 96-97 in the revised version of the manuscript.

Point 7(M&M): Section 2.3.1, and 2.3.2, are not clear? How?

Response 7: We revised section 2.3.1. and 2.3.2 for providing more information on temperature/pH/color measurements. Please see line 104-114 in the revised version of the manuscript.

Point 8(M&M): What is difference between 2.3.1 and 2.3.1. regarding the ph measurement?

Response 8: Section 2.3.1 was structured to explain the methodological description on the pH measurement of intact muscle, whereas section 2.4.1 was for the pH measurement of ground and salted chicken breasts.

Point 9(M&M): Section 2.4.2 is not clear

Response 9: The procedure of cooking loss is one of the common methods to evaluate the water-holding capacity during thermal treatment. Since the methodological procedure is quite general, it is hard to understand what the unclear point is here. If it is still insufficient, please mention it again in more detail.

Point 10(M&M): The statistical analysis is not clear

Response 10: We rephrased the statistical analysis section for readers’ better understanding. Please see line 173-180 in the revised version of the manuscript.

Point 11(M&M): Table 1, Table 2 are not clear ( the statistical results are not clearly presented and discussing the results section)

Response 11: We partially revised Table 1 and 2 for better understanding. Please see Table 1 and 2 in the revised version of the manuscript.

Point 12(L216): Can the pH inactivates the enzymes? Or inhibit? What about effect of lower ph on the enzyme activities?

Response 12: Pre-rigor salting could inactivate the enzyme related to glycolysis under high ionic strength. Regarding this point, we cited relevant references (reference 5 and 22).

Point 13(Table 3): Table 3 is not discussed well.

Response 13: We double-checked all results described in results and discussion section. If it is still insufficient, please mention it again in more detail.

Point 14(L241-243): what is the hypothesis of this paper? If other studies already proved the salt type, does not have

Response 14: The hypothesis of this study is that the salting effects of KCl and NaCl on technological properties of chicken breasts may be different at the same ionic strength, and their salting effect would be affected by rigor status of raw meat.